# Dietary Interventions for Cancer Prevention: An Update to ACS International Guidelines

**DOI:** 10.3390/nu16172897

**Published:** 2024-08-29

**Authors:** Álvaro Torres, Francisca Quintanilla, Esteban Barnafi, César Sánchez, Francisco Acevedo, Benjamín Walbaum, Tomás Merino

**Affiliations:** 1School of Medicine, Faculty of Medicine, Pontificia Universidad Católica de Chile, Santiago 8331150, Chile; astorres1@uc.cl (Á.T.); fran.quintanilla@uc.cl (F.Q.); estebanandres@barnafi.com (E.B.); 2Department of Hematology-Oncology, Faculty of Medicine, Pontificia Universidad Católica de Chile, Santiago 8330077, Chile; csanchez@med.puc.cl (C.S.); fnaceved@uc.cl (F.A.); bvwalbau@uc.cl (B.W.); 3Cancer Center UC, Red de Salud Christus-UC, Santiago 8330032, Chile

**Keywords:** diet, nutrition, cancer, risk, prevention, Mediterranean diet, dairy, vitamin D, diet quality, carbohydrates

## Abstract

Cancer, the second leading cause of death worldwide, demands the identification of modifiable risk factors to optimize its prevention. Diet has emerged as a pivotal focus in current research efforts. This literature review aims to enhance the ACS guidelines on diet and cancer by integrating the latest findings and addressing unresolved questions. The methodology involved an advanced PubMed search with specific filters relevant to the research topic. Topics covered include time-restricted diet, diet quality, acid load, counseling, exercise and diet combination, Mediterranean diet, vegetarian and pescetarian diets, weight loss, dairy consumption, coffee and tea, iron, carbohydrates, meat, fruits and vegetables, heavy metals, micronutrients, and phytoestrogens. The review highlights the benefits of the Mediterranean diet in reducing cancer risk. Adherence to overnight fasting or carbohydrate consumption may contribute to cancer prevention, but excessive fasting may harm patients’ quality of life. A vegetarian/pescetarian diet is associated with lower risks of general and colorectal cancer compared to a carnivorous diet. High heme and total iron intake are linked to increased lung cancer risk, while phytoestrogen intake is associated with reduced risk. Coffee and tea have a neutral impact on cancer risk. Finally, the roles of several preventive micronutrients and carcinogenic heavy metals are discussed.

## 1. Introduction

Traditionally, eating habits have been conceived as a matter of balancing caloric intake and the types of macronutrients consumed [1,2,3]. This vision is centered on the role of diet in maintaining body weight and meeting the body’s physiological needs. However, contemporary research has brought about a shift aiming to understand the influence of diet in health and disease. Indeed, diet is now widely recognized not only as a determinant of physical well-being but also as a modifiable risk factor linked to the prevalence and prognosis of various health conditions [4,5,6]. In fact, dietary patterns are associated with chronic diseases such as cardiovascular disorders [4], obesity [5], type 2 diabetes, and cancer [6]. Consequently, food choices are no longer considered isolated decisions in terms of calories or macronutrients, but rather as integral components of complex interactions between diet and general overall health.

Worldwide, cancer remains one of the leading causes of death, with an estimated 19.3 million new cases and nearly 10.0 million deaths only in 2020 [7]. Among the factors that contribute to the global burden of cancer, diet has been proposed as a modifiable risk factor with potential roles in cancer prevention, recurrence, and overall survival. Diet can also improve other health-related outcomes, such as survival and quality of life [8,9,10,11].

In recent decades, the association between diet and cancer has sparked the interest of investigators. In 2020, the American Cancer Society (ACS) published diet and physical activity guidelines for cancer prevention [12]. Summaries of the ACS’s recommendations are shown in Table 1 and Table 2.

This review aims to deliver an update on these recommendations by summarizing novel evidence and highlighting areas that require further investigation. Some of the discussed topics include associations between cancer risk and time-restricted eating, consumption of dairy, vegetarian and pescatarian diets, coffee and tea, iron, meat, fruits and vegetables, acid charge, vitamin D, and phytoestrogens. We also include the potential role of professional advice, dietary patterns (such as the Mediterranean diet (MD)), and diet quality in cancer risk [13]. Diet quality levels are based on the application of specific indexes, outlined in Table 3, which have shown a moderate predictive capacity for the incidence of chronic diseases and other health determinants [14].

This literature review aims to update known knowledge with new research published and highlight areas that still require further study. The main points of discussion will be time-restricted feeding, dairy, vegetarian and pescetarian diets, coffee and tea, iron, meat, fruits and vegetables, acid charge, vitamin D, phytoestrogens, professional advice, dietary patterns (such as Mediterranean diet (MD)), and diet quality, which refers to both the quantity of nutrients and the absorption of specific nutrients from food [14]. These indexes have shown a moderate capacity to predict the incidence of chronic diseases and other health determinants, highlighting their relevance [15] to prostate, breast, lung, colon, and the overall risk of cancer.

## 2. Materials and Methods

### 2.1. Search Strategy

The initial phase of our study (visualized in Figure 1) began with the development of an effective search strategy to capture as much relevant literature as possible. This was achieved through a combination of general and advanced search query parameters: “‘cancer’ AND ‘nutrition’” and, alternatively, “‘cancer’ AND ‘diet’”, along with Boolean operators (’AND’, ’OR’, ’NOT’) to expand or limit the search as needed, and specific filters (such as the selection of randomized clinical trials and a publication period within 6 years). This was done on several academic databases, including, but not limited to, Google Scholar, PubMed, and ScienceDirect. Within this set of resources, priority was given to PubMed Advanced Search Results as our main source of information.

### 2.2. Screening and Selection

The search results underwent a screening and selection process. First, we selected an internationally renowned and relevant medical association’s guideline and used it as the basis for our review (the ACS guideline for diet and physical activity for cancer prevention and cancer survivors [12,13]). After this we found studies published shortly before and after our selected guideline. Only studies published between 2018 and 2024 were considered. These articles were revised to assess their relevance to the research question. Articles that were reviews of other studies were discarded. Exceptionally, a review was considered to discuss micronutrients in more depth, which is cited in the Section 3. The remaining articles were reviewed by our team to ensure they provided new insight into the selected review topic, or else were entirely new topics not previously covered. The ACS guideline references were reviewed with Google Scholar to ensure the literature found was not previously cited. Thus, studies that were not cited by any guideline are presented in the Section 3.

### 2.3. Synthesis and Reference Verification

Once we selected the final group of articles, we implemented two measures to ensure the reliability and robustness of our paper. First, as already mentioned, we verified the citations of each article using Google Scholar and its advanced search function, making sure that none of the selected articles had been previously cited in an earlier guideline. Subsequently, we proceeded to draft a synthesis of each article with the purpose of more effectively organizing the conclusions obtained from the selected research. With all of this, we had 24 original articles left that had not been referenced and therefore could be considered as new or novel.

## 3. Results

A total of 33 studies (summarized in Table 4) were included in our analysis. Most relevant conclusions for these studies are shown below.

### 3.1. Time-Restricted Eating

A case-control study [16] suggested that prolonged nighttime fasting and an early breakfast could be associated with a lower risk of prostate cancer.

### 3.2. Mediterranean Diet

A case-control study conducted in Italy [17] with over 3000 patients found that higher adherence to the MD reduced the risk of breast cancer. A Polish study suggested that pro-healthy dietary patterns, including the Mediterranean pattern, may favor a lower risk of lung cancer in moderate smokers, although this was not confirmed in heavy smokers [18]. However, a dietary intervention involving 115 healthy individuals [19] found that the Mediterranean diet did not reduce trimethylamine N-oxide (TMAO) concentrations, a compound that has been identified as a risk factor for numerous cardiovascular diseases and colon cancer [20] The above shows a lack of knowledge of the mechanisms of how this diet works.

### 3.3. Dairy

Individual-level data from over 1 million women show that there is no clear association between the consumption of specific dairy products, dietary calcium, and total calcium, and the risk of overall breast cancer [21].

### 3.4. Vegetarians and Pescetarians

Vegetarians and pescetarians had a lower risk of overall cancer and colon cancer compared to carnivores [22].

### 3.5. Coffee and Tea

No evidence of an association between total coffee or tea consumption, with or without caffeine, and the risk of overall prostate cancer or cancer by stage, grade, or mortality in this large cohort exists [23].

### 3.6. Iron

Heme iron intake was moderately associated with the risk of lung cancer, and non-heme iron showed an inverse association that was corrected after adjusting for smoking history [24]. The genetic level of iron was not causally associated with the risk of lung cancer [25].

By contrast, a Uruguayan study [26] found that total iron, animal, heme, and non-heme animal iron were positively associated with lung cancer, while plant and non-heme fraction in total iron were inversely associated with lung cancer incidence. Furthermore, stratified analyses showed higher ORs for heavy smokers and heavy maté drinkers.

### 3.7. Carbohydrates

A Japanese study [27], included in the Korean guidelines for gastric cancer [28], concluded that low-carbohydrate diets are associated with a higher risk of overall incidence of colorectal and lung cancer but reduce the risk of gastric cancer.

### 3.8. Diet Quality

A study based on data from the Women’s Health Initiative Observational Study concluded that, although overall diet quality was not associated with lung cancer in general, a high-quality diet showed an inverse relationship with the incidence of squamous cell lung cancer [29]. These results were independent of the patient’s age and smoking habit. These findings are similar to another study conducted in a multiethnic group, which also indicated that high-quality diets are associated with a lower risk of lung cancer, especially the squamous cell subtype [30]. On the other hand, another study with Iranian adults found that a proinflammatory diet (using the Dietary Inflammatory Index) increased the likelihood of lung cancer in adults, mainly in men [31]. The Golestan Cohort Study concluded that higher DASH-Fung and HEI-2015, each respectively, were inversely associated with lung cancer risk [32].

A Finnish cohort [33] had a 33% decrease in postmenopausal breast cancer risk for those with the highest diet quality compared with those with the lowest quality according to the Nordic diet; (95% CI 0.48–1.01). While diet quality measured by mMEDI and mAHEI were not associated with postmenopausal breast cancer risk.

### 3.9. Meat

A large prospective cohort study conducted in the UK [34] with over 400,000 patients demonstrated that a high intake of red and processed meat was associated with an increased risk of lung cancer.

### 3.10. Fruits and Vegetables

The same large UK study demonstrated that the consumption of fruits, vegetables, breakfast cereals, and dietary fiber were inversely associated with the risk of lung cancer [34]. This study [35] supports cancer prevention strategies with a diet rich in vegetables, fruits, fish, and whole grains.

### 3.11. Acid Charge

The net endogenous acid production (NEAP) score, directly associated with meat consumption and inversely associated with vegetable consumption, was significantly related to an increased risk of all types of lung cancer, except for small cell cancers [36].

### 3.12. Phytoestrogens

The Prostate, Lung, Colorectal, and Ovarian Cancer Screening Trial (PLCO) [37] found that the intake of phytoestrogens (PE) was associated with a reduced risk of lung cancer.

### 3.13. Importance of Professional Advice

The Oslo diet and the anti-smoking study [38] showed that 5-year counseling for a healthy lifestyle did not reduce overall long-term cancer risk. However, in the first 25 years, counseling reduced the risk of relevant cancer types in overweight/obese subjects and smokers.

### 3.14. Alternative Diet Advice

A large Canadian study [35] showed that high adherence to a prudent eating pattern (characterized by a high intake of vegetables, fruits, and lean meat from fish and other seafood) was associated with a lower risk of lung cancer.

In 2019, the EAT-Lancet commission proposed a universal dietary pattern that was analyzed in 2022 [39]. It was found that adherence to the EAT-Lancet diet was not associated with a significantly lower risk of cancer, and only among those with low alcohol consumption was a decrease in cancer risk observed.

The influence of the paleolithic diet, centered on vegetables, fruits, nuts, fish, and lean meat [40], revealed that women with a high adherence to this diet experienced a 17% lower risk of developing breast cancer. This benefit was sustained when analyzing the different subtypes of breast cancer, evidencing a consistent pattern of association.

### 3.15. Heavy Metals

A recent review shows an association between exposure to heavy metals and the risk of developing cancer. Elevated levels of metals such as copper, lead, and zinc are associated with an increased risk of breast cancer [41]. In particular, a marginal positive relationship has been identified between dietary intake of cadmium and breast cancer, although this relationship was not observed when exposure was assessed through urinary cadmium excretion [42]. In addition, it has been reported that edible mushrooms may be a significant source of toxic heavy metals, which could have a negative impact on long-term health [43].

In terms of dietary interventions, some studies have investigated strategies to mitigate heavy metal exposure through diet. For example, daily rice bran intake for six months in Nicaraguan infants did not result in significant differences in the levels of trace elements and heavy metals in serum and feces, suggesting controlled exposure [44]. In addition, the use of organic residue treatments in crop soils has been shown to be effective in reducing heavy metal levels in corn kernels, which could offer health benefits by reducing exposure to these contaminants [45].

### 3.16. Micronutrients

Low levels of zinc (Zn) in women with the BRCA1 mutation is weakly associated with an increased risk of ovarian cancer, although there was no similar association with breast cancer [46]. A high zinc/copper (Zn/Cu) ratio could reduce cancer risk in this same population [47], suggesting the importance of optimizing these minerals for cancer prevention in BRCA1 carriers [47].

In prostate cancer patients, low levels of selenium (Se) and Zn are associated with shorter 5-year survival, reinforcing the importance of maintaining adequate levels of both elements to improve prognosis [48]. In addition, high levels of Zn and Se are associated with longer survival in several types of cancer, whereas high levels of copper (Cu) may increase mortality in breast, prostate, lung, and laryngeal cancers [49].

As for vitamins, vitamin C can enhance the ability of NK cells to kill tumor cells [50,51]. Vitamin D, identified as key in the immune system because of its receptor on macrophages, dendritic cells, and activated T and B lymphocytes, plays an important role in immune modulation [50,52]. However, a Finnish study showed that vitamin D3 supplementation does not reduce the incidence of cancer in individuals with adequate levels of vitamin D3 [53].

On the other hand, vitamin E, a potent antioxidant, protects cell membranes and enhances immunity [50,54]. Retinoic acid (RA), derived from vitamin A, regulates cytokine release and could prevent cancer by influencing Th1/Th2 immune pathway differentiation [50,55].

## 4. Discussion

In line with the ACS recommendations [12] that emphasize the importance of a healthy diet in the prevention of disease, several studies highlight the protective role of the MD, along with abundant consumption of fruits and vegetables, against cancer [17,34,35]. Studies also point out the potentially harmful effects of excessive red meat consumption, which could be at least partly explained by a higher acid load, as evidenced by the high NEAP Score [36]. On the other hand, the fact that lower levels of TMAO do not correlate with MD adherence [19] warrants further investigation into other potential physiological mechanisms, beyond oxidative stress, that could explain its protective role.

Although some studies have attributed specific benefits to certain diets, such as the prudent eating pattern or the paleolithic diet [35,40], the scientific evidence that supports the cancer prevention properties of the MD is overwhelming. Therefore, adherence to this dietary pattern as a cancer preventive strategy should be encouraged.

Objective assessments of diet quality using validated indices, such as the HEI-2015, AHEI-2010, aMED Score, DII, and DASH Score are valuable tools for the prevention of chronic, cardiovascular, and neoplastic diseases. Therefore, in our opinion, these indices should be included in specialized clinical guidelines, allowing a personalized quantification of risk by health professionals according to patients’ dietary habits.

Regarding time-restricted eating strategies, we consider the evidence on its effects and potential risks as still uncertain; therefore, more studies are required to incorporate these strategies into specialized guidelines. In contrast, the evidence that supports the beneficial role of phytoestrogens in cancer prevention is more consistent, particularly after the multicenter cancer screening study by Wang et al. that involved >100,000 individuals [37], suggesting that the consumption of phytoestrogens deserves to be considered in specialized guidelines.

Although previous reports by the NCCN suggested that higher consumption of cows’ milk was associated with an increased BC risk [56], a more recent publication by You Wu [21] contradicts this report, suggesting that additional research is required to confirm or discard this association. Similarly, many studies have demonstrated a lack of significant effects in cancer prevention of certain foods like coffee and tea [10,23]. Considering this information, it would be useful to dedicate a formal section in specialized guidelines emphasizing the lack of effect on cancer risk of these foods. For example, a study by Hong Qin [25], based on a European database, determined a lack of association between genetic iron levels and lung cancer risk. Interestingly, this study was based on a specific population, which opens an important field of research regarding the influence of genetic factors in specific ethnic groups.

Another cancer preventive recommendation by the ACS is to avoid highly refined carbohydrates [12]. While certain carbohydrates may be associated with carcinogenesis, others are linked to protective effects, suggesting more research is needed to determine the specific amounts that balance cancer risk versus benefits.

The analyzed literature suggests that professional counseling should continue lifelong for the lifestyle of both patients and healthy subjects, and should not be limited to specific interventions.

As opposed to the MD and its benefits, exposure to heavy metals poses a significant cancer risk factor, particularly for BC. Therefore, periodical monitoring of copper, lead, and zinc levels, along with a strict diet avoiding the exposure to these metals, should be recommended to the general population, especially individuals at a higher for BC [41]. Although the association between BC and dietary intake of cadmium is marginal and dependent on assessment methods, these findings warrant further investigation into the potential underlying mechanism(s) [42]. Evidence also suggests that certain foods, such as mushrooms, can be a significant source of toxic heavy metals, highlighting the importance of careful food selection and dietary interventions [43]. In this regard, interventions such as daily rice bran did not show a significant impact on heavy metal levels in the body, suggesting that some dietary strategies may be insufficient to reduce exposure [44]. In contrast, agricultural practices that reduce the concentration of heavy metals in food, such as the use of organic residues in soils, offer a promising intervention that could reduce cancer risk by limiting dietary exposure to these contaminants [45]. Evidently, this is another topic that demands further research in the near future.

Regarding micronutrients, such as vitamins, minerals, and antioxidants, these play a vital role in the maintenance of cellular health and the reduction of oxidative stress. Although preliminary studies did not find an association between serum Zn levels and cancer risk in BRCA1 mutation carriers [46], the Zn/Cu ratio may be a biomarker in this subset of women [47]. Therefore, it is necessary to consider the optimization of serum levels of both Zn and Cu to have a more useful reference in clinical practice. In this context, studies demonstrate the impact of Se and Zn levels on the survival of prostate cancer patients [48]. Furthermore, a trial seeking to confirm these claims by systematically measuring serum levels of these microelements is currently underway.

The results of our literature search confirm and extend our current understanding of the relationship between vitamins and cancer risk, emphasizing their role as immune modulators that also protect our DNA. Within this context, the activating effect of vitamin C on NK cell function and its inhibitory effect on tumor cell migration suggests a potential role in the prevention of cancer spread and metastasis. Obviously, its potential clinical use in the future will require further research to optimize doses and conditions [50,51]. As for vitamin D, although its receptor has been identified on various immune cells, suggesting a broad immunomodulatory role, a Finnish study indicates that additional supplementation in individuals with adequate vitamin D levels does not reduce cancer incidence, suggesting the benefit is limited to deficient populations, which also highlights the importance of assessing basal levels prior to supplementation measures or recommendations [50,52,54].

Besides its potent antioxidant activity, vitamin E enhances T-cell function and inhibits COX-2, which may act as an immunomodulatory mechanism, stimulating antitumor immunity. Unfortunately, the evidence for this mechanism is mostly based on experimental studies, and the confirmation of potential clinical benefits in humans demands further investigation. In this regard, retinoic acid, an active form of vitamin A, has a dual effect in being both proinflammatory and anti-inflammatory, reflecting the complexity of immune modulation. Its ability to influence Th1/Th2 pathway differentiation and regulate cytokine production suggests a therapeutic potential in modulating specific immune responses in the context of cancer. However, its role in cancer prevention and treatment should be approached with caution, as the effects may vary according to the inflammatory context and the type of neoplasm. Overall, these findings point to more personalized vitamin supplementation, balancing benefits and risks for every individual.

## 5. Limitations and Future Direction of Work

Our literature search had several limitations. Firstly, selecting studies from the last six years may introduce temporal bias. Secondly, the search was conducted on PubMed, Google Scholar, and ScienceDirect, potentially omitting relevant studies published in other databases. Thirdly, our review was based on published literature, possibly omitting studies with negative or non-significant results. Finally, the multifactorial nature of the relationship between diet and cancer development makes it challenging to fully address the complexity of this association, leaving room for incomplete exploration of the interaction between various genetic and environmental factors.

## 6. Conclusions

Dietary habits are not only a key determinant of physical well-being, but also a modifiable risk factor directly associated with chronic diseases such as obesity, type 2 diabetes, hypertension, and cancer. Our findings support the ACS recommendations and highlight the need for deeper research in specific areas, such as time-restricted diets, structured physical activity programs, the impact of alternative diets (not MD) on cancer risk, and the effect of unprocessed carbohydrates.

## Figures and Tables

**Figure 1 nutrients-16-02897-f001:**
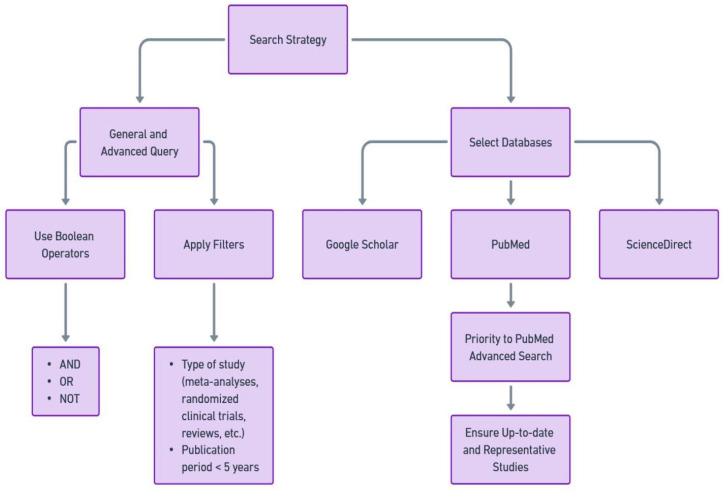
Search strategy employed to find publications involved in our area of interest.

**Table 1 nutrients-16-02897-t001:** Modified from ACS recommendations on breast, colorectal, lung and prostate cancer [12,13].

Breast	Increases Risk	Weight gain during adult life
		Excess body fatness
		Alcohol consumption
		Weight loss
	Decreases Risk	Physical activity (moderate or vigorous)
		Diets rich in plant food
		Diets low in animal products
		Diets low in refined carbohydrates
		Mediterranean diet
	May Decrease Risk	Non-starchy vegetables
Colorectal	Increases Risk	Excess body fatness
		Alcohol consumption
		Processed and red meat
		Low circulating levels of vitamin D
	Decreases Risk	Physical activity (moderate or vigorous) (only colon)
		Whole grains, higher fiber, and less added sugar
		Diets higher in calcium/calcium-rich dairy foods
		Reducing sedentary behavior (only colon)
		Non-starchy vegetables
		Vegetables rich in carotenoids
		Supplemental calcium
Lung	Increase Risk	High-dose beta-carotene supplementation
		Exposure to asbestos
		Alcohol consumption
		Processed and red meat
	Decreases Risk	Physical activity (moderate or vigorous)
		Reducing sedentary behavior (only colon)
		Non-starchy vegetables
		Whole fruits, especially those high in vitamin C for smokers
Prostate	Increases Risk	Excess body fatness
	May Increase Risk	Higher consumption of dairy products and calcium

**Table 2 nutrients-16-02897-t002:** General health recommendations [12,13].

Overall Recommendations	Achieve and maintain healthy body weight	
	Be physically active	
	Healthy eating pattern	Foods high in nutrients
		Vegetables
		Fruits
		Whole grains
		Legumes
	Limit or not include intake of the following	Red and processed meats
		Sugar-sweetened beverages
		Highly processed foods
		Refined grain products
	Not drink alcohol	

**Table 3 nutrients-16-02897-t003:** Assessment indexes that can be used to assess diet quality [15].

Diet Quality Assessment Indexes
Healthy Eating Index-2015 (HEI-2015)
Alternative Healthy Eating Index-2010 (AHEI-2010)
Alternative Mediterranean Diet (aMED)
Dietary Approaches to Stop Hypertension (DASH) score
Dietary Inflammatory Index (DII^®^) score

**Table 4 nutrients-16-02897-t004:** Summary of studies on dietary interventions and cancer risk.

Author	Year	Population Origin	Type	*n*	Population	Intervention	Cancer	Key Findings
Palomar-Cros	2021	Spain	MCC	607 + 848	PC + C	Nighttime fasting and breakfast	PC	Lower risk
Turati	2018	Italy/Switzerland	CC	3034 + 3392	BC + C	Mediterranean diet	BC	Reduced risk
Hawrysz	2020	Poland	CC	187 + 252	LC + NC	Dietary pattern	LC	Lower risk in moderate smokers
Griffin	2019	North America	ND	115	Healthy	Mediterranean diet	CC	No change in fasting TMAO
Wu	2021	WW	ES	37,861	BC	Dairy + calcium	BC	No association was observed
Parra-Soto	2022	UK	PC + Ma	409,110	UK Biobank	Dietary pattern	OC	Lower risk
Sen	2019	Europe	PS	142,196	14y FU	Coffee and tea	PC	No association was observed
Ward	2019	Europe	PS	478,590	6.4y FU	Heme iron	LC	Association was found
Qin	2022	Europe	OS	48,972	LC	Genetic iron status	LC	No association was observed
Ronco	2019	Uruguay	CC	843 + 1466	LC + C	Iron intake	LC	Association was found
Cai	2022	Japan	PS	90,171	17y FU	Low- carbohydrate diet	OC	Association was found
Myneni	2021	WW	PS	86,090	17y FU	Diet quality	LC	Some associations were found
Park	2021	Multiethnic	PS	179,318	17.5y FU	Diet quality	LC	Lower risk
Sadeghi	2022	Middle East	CC	140 + 140	LC + C	Diet quality	LC	Increases risk
Wang	2021	Middle East	PS	48,421	12y FU	Diet quality	LC	Lower risk
Männistö	2021	Finnish	PS	6374	10y FU	Diet quality	BC	Lower risk
Wei	2021	UK	PS	416,588	7y FU	Diet	LC	Lower risk
Willemsen	2022	Canada	PS	26,462	13.3y FU	Dietary pattern	OC	Lower risk
Ronco	2021	Uruguay	CC	843 + 1466	LC + C	Acid load	LC	Association was found
Virtanen	2022	Finnish	RCT	2495	5y FU	Vitamin D	OC	No association was observed
Wang	2021	USA	PS	101,712	12.2y FU	Phytoestrogen	LC	Lower risk
Botteri	2018	Norway	RCT	595 + 621	I + C	Counseling	OC	Some associations were found
Berthy	2022	France	PS	62,382	8.1y FU	EAT-Lancet Diet	OC	Some associations were found
Shah	2023	France	PS	65,574	20y FU	Paleolithic diet	BC	Lower risk
Filippini	2020	USA/Japan/Europe	Ma			Cadmium exposure	BC	Some associations were found
Dowlati	2021	Multiethnic	Ma		Healthy	Heavy metals	NC	Some associations were found
Zambrana	2021	Nicaragua	PS	47	6m FU	Heavy metals	NC	No association was observed
Ma	2020	China		-	-	Heavy metals	NC	
Liu	2022	Multiethnic	Ma	4151	BC + C	Heavy metals	BC	Some associations were found
Lubinski	2023	Poland	PS	1475	9.8y FU	Micronutrients	OC	Reduced risk
Matuszczak	2024		PS	989	7.5y FU	Micronutrients	BC/OvC	No association was observed
Matuszczak	2024	Poland	PS	989	7.5y FU	Micronutrients	BC/OvC	Reduced risk
Pietrzak	2024	Poland	PS	338	6y FU	Micronutrients	PC	Some associations were found

MCC = Multicase control; CC = case-control; ND = no data; ES = epidemiological study; PS = prospective study, Ma = meta-analysis; OS = observational study; RCT = randomized controlled trial; PC = prostate cancer; BC = breast cancer; LC = lung cancer; NC = no cancer; C = controls; FU = follow-up; CC = colon cancer; OC = overall cancer; OvC = ovarian cancer; I = intervention.

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
