# Peer review of "Dietary Interventions for Cancer Prevention: An Update to ACS International Guidelines"

_nutrients, 2024, doi:10.3390/nu16172897_

Round 1

Reviewer 1 Report

Comments and Suggestions for Authors

Review of "Dietary Interventions for Cancer Prevention: An Update to ACS International Guidelines"

The article "Dietary Interventions for Cancer Prevention: An Update to ACS International Guidelines" provides a comprehensive overview of current dietary recommendations aimed at reducing the risk of cancer. While the article covers a broad range of dietary strategies and their implications, it has notable gaps, particularly in the areas of referencing and the discussion of micronutrients and heavy metals. This review will address these shortcomings and suggest enhancements based on recent research.

Missing References in Tables

Tables 1 and 2 in the article lack proper references, which undermines the credibility and reliability of the data presented. Including references is essential for validating the information and providing readers with sources for further reading. It is recommended to update these tables with appropriate citations to support the data and strengthen the overall impact of the article.

Impact of Diet on Micronutrients and Heavy Metals

One significant omission in the article is the discussion of how dietary interventions affect the levels of micronutrients and heavy metals in the blood or serum, which are critical factors in cancer prevention. Micronutrients such as vitamins, minerals, and antioxidants play a vital role in maintaining cellular health and preventing oxidative stress, which can lead to cancer. Similarly, the presence of heavy metals in the body can contribute to carcinogenesis.

Micronutrients

Recent studies have highlighted the importance of various micronutrients in cancer prevention. For example, articles: DOI:10.3390/antiox13070841, DOI:10.3390/antiox13050609, DOI:10.3390/nu16040527, 10.3390/nu15112611. 

Another study by Zhou et al. (2023) discusses the impact of specific vitamins and minerals, including vitamins A, C, D, and E, selenium, and zinc, in modulating immune responses and protecting against DNA damage. These micronutrients are essential for maintaining the integrity of cellular processes and preventing mutations that could lead to cancer.

Heavy Metals

The influence of dietary interventions on the levels of heavy metals in the body is another critical aspect that the original article does not address. Heavy metals such as lead, cadmium, mercury, and arsenic are known carcinogens that can be found in contaminated food and water. es.

Recommendations for Article Enhancement

Include References in Tables: Update Tables 1 and 2 with appropriate citations to enhance the credibility of the data presented.

Discuss Micronutrient Impact: Integrate findings from recent studies on the role of antioxidants and essential vitamins and minerals in cancer prevention. Highlight the importance of a diet rich in fruits, vegetables, and whole grains.

Address Heavy Metals: Incorporate a discussion on the impact of heavy metals on cancer risk and the importance of reducing exposure through dietary choices. Emphasize the role of a balanced diet in mitigating the effects of these toxic elements.

Holistic Approach: Encourage a holistic approach to cancer prevention that considers both the benefits of essential nutrients and the risks posed by heavy metals. This will provide readers with a more comprehensive understanding of how dietary choices can influence cancer risk.

By addressing these gaps and integrating recent research, the article can provide a more thorough and informative guide on dietary interventions for cancer prevention.

Author Response

Comments 1: [Tables 1 and 2 in the article lack proper references, which undermines the credibility and reliability of the data presented. Including references is essential for validating the information and providing readers with sources for further reading. It is recommended to update these tables with appropriate citations to support the data and strengthen the overall impact of the article.]

Response 1: [The references for Tables 1 and 2 are already included in the bibliography, specifically in citation number 12 and 13 on lines 393-400.]

Comments 2: [One significant omission in the article is the discussion of how dietary interventions affect the levels of micronutrients and heavy metals in the blood or serum, which are critical factors in cancer prevention. Micronutrients such as vitamins, minerals, and antioxidants play a vital role in maintaining cellular health and preventing oxidative stress, which can lead to cancer. Similarly, the presence of heavy metals in the body can contribute to carcinogenesis.]

Response 2: [Both the Results and Discussion sections have been updated to include the requested topics on micronutrients and heavy metals. It was addressed from their relevance to dietary interventions that current evidence suggests might be useful.]

Comments 3: [Recent studies have highlighted the importance of various micronutrients in cancer prevention. For example, articles: DOI:10.3390/antiox13070841, DOI:10.3390/antiox13050609, DOI:10.3390/nu16040527, 10.3390/nu15112611. 

Another study by Zhou et al. (2023) discusses the impact of specific vitamins and minerals, including vitamins A, C, D, and E, selenium, and zinc, in modulating immune responses and protecting against DNA damage. These micronutrients are essential for maintaining the integrity of cellular processes and preventing mutations that could lead to cancer.

The influence of dietary interventions on the levels of heavy metals in the body is another critical aspect that the original article does not address. Heavy metals such as lead, cadmium, mercury, and arsenic are known carcinogens that can be found in contaminated food and water]

Response 3: [All articles suggested to complement the information were incorporated in the corresponding sections, except the work of Zhou et al. (2023), which could not be identified due to the incomplete description provided. However, the team addressed the suggested topic, delving into the role of certain vitamins and micronutrients in the regulation of immune response and DNA protection. The references used for this topic can be found in citations 50 to 55 (lines 493-506). If you consider it necessary, we would appreciate if you could share with us the specific DOI of Zhou's work to integrate it into the text.]

Comments 4: [Holistic Approach: Encourage a holistic approach to cancer prevention that considers both the benefits of essential nutrients and the risks posed by heavy metals. This will provide readers with a more comprehensive understanding of how dietary choices can influence cancer risk.]

Response 4: [A more accentuated promotion of a holistic approach to dietary interventions for cancer prevention, with emphasis on a balanced diet that considers adequate levels of micronutrients and heavy metals, was added to the discussion. Besides, an exhaustive revision of the English language was made in the text, integrating the ACS recommendations and trying to present the clinical suggestions in a critical and transparent manner.]

Reviewer 2 Report

Comments and Suggestions for Authors

The American Cancer Society (ACS) publishes the Diet and Physical Activity Guideline to influence dietary and physical activity patterns among Americans. This guideline is developed by a national panel of experts in cancer research, prevention, epidemiology, public health, and policy, reflecting the most current scientific evidence related to dietary and activity patterns and cancer risk. This article has some reference value. However, the following details require attention:

As a review article to be published in this Journal, it is slightly lacking. The author can combine the two articles (DOI: 10.3322/caac.21591 and DOI: 10.3322/caac.21719) to improve the readability and innovation of the article through systematic elaboration and comprehensive analysis.

Comments on the Quality of English Language

 Minor editing of English language required.

Author Response

Comments 1: [ Minor editing of English language required.]

Response 1: [The text was re-edited and refined in English, integrating more effectively the recommendations previously established by the ACS. Efforts were made to promote in a critical and transparent manner recommendations that may be clinically significant.]

Comments 2: [As a review article to be published in this Journal, it is slightly lacking. The author can combine the two articles (DOI: 10.3322/caac.21591 and DOI: 10.3322/caac.21719) to improve the readability and innovation of the article through systematic elaboration and comprehensive analysis.]

Response 2: [The DOIs indicated were considered from the beginning of the drafting of the manuscript and are cited as references 12 and 13 in the text. However, it is important to note that the focus of this work is particularly on cancer prevention, which led us to pay particular attention to the ACS guidelines.

To complement the information presented in the guides with the recommendations that this revision leaves in question, the following changes were made:

  1. Tables 1 and 2: The reference used for the construction of Tables 1 and 2 was added to provide greater credibility and solid support for an adequate critical analysis.
  2. Updated sections: Both the Results and Discussion sections were updated to include the “micronutrients” and “heavy metals” topics. The “Vitamin D” section was integrated into the micronutrients section to provide a more complete analysis of the available evidence. The specific relevance of these topics was addressed, as well as dietary interventions that recent studies suggest might be useful.
  3. Holistic approach: The promotion of a holistic approach to dietary interventions for cancer prevention was also reinforced in the discussion, highlighting the importance of a balanced diet that considers adequate levels of the components outlined in the paper.]

Round 2

Reviewer 2 Report

Comments and Suggestions for Authors

The author has revised the manuscript and given reasonable explanation, I think the revised manuscript can be accepted now.